

# SNARE-CNN: a 2D convolutional neural network architecture to identify SNARE proteins from high-throughput sequencing data

Nguyen Quoc Khanh Le[1] and Van-Nui Nguyen[2]

[1] School of Humanities, Nanyang Technological University, Singapore
[2] University of Information and Communication Technology, Thai Nguyen University, Thai Nguyen, Vietnam

## ABSTRACT

Deep learning has been increasingly and widely used to solve numerous problems in various fields with state-of-the-art performance. It can also be applied in bioinformatics to reduce the requirement for feature extraction and reach high performance. This study attempts to use deep learning to predict SNARE proteins, which is one of the most vital molecular functions in life science. A functional loss of SNARE proteins has been implicated in a variety of human diseases (e.g., neurodegenerative, mental illness, cancer, and so on). Therefore, creating a precise model to identify their functions is a crucial problem for understanding these diseases, and designing the drug targets. Our SNARE-CNN model which uses two-dimensional convolutional neural networks and position-specific scoring matrix profiles could identify SNARE proteins with achieved sensitivity of 76.6%, specificity of 93.5%, accuracy of 89.7%, and MCC of 0.7 in cross-validation dataset. We also evaluate the performance of our model via an independent dataset and the result shows that we are able to solve the overfitting problem. Compared with other state-of-the-art methods, this approach achieved significant improvement in all of the metrics. Throughout the proposed study, we provide an effective model for identifying SNARE proteins and a basis for further research that can apply deep learning in bioinformatics, especially in protein function prediction. SNARE-CNN are freely available at https://github.com/khanhlee/snare-cnn.

## INTRODUCTION

Deep learning is an advanced machine learning and artificial intelligent technique to learn the representative data with multiple layers of neural networks (*LeCun, Bengio & Hinton, 2015*). Numerous difficult problems have been solved with deep learning, e.g., speech recognition, visual object recognition, object detection. The advantages of deep learning are: (1) significantly outperforms other solutions in multiple domains, (2) reduces the requirement for feature extraction and time consumption with the use of graphic processing units (GPUs), and (3) easily adapts to a new problem. Deep neural network

Corresponding author
Nguyen Quoc Khanh Le,
khanhle@ntu.edu.sg,
khanhlee87@gmail.com

models often achieve better performance compared to shallow networks, especially in most of problems with big data. Therefore, deep learning becomes popular and attracts numerous huge companies establishing their directions in this field in recent years. Nowadays, much progress towards deep learning has been made using different deep neural network architectures. A number of studies showed that using deep learning can enhance results in various fields, e.g., prediction of cervical cancer diagnosis (*Fernandes et al., 2018*), piRNA (*Wang, Hoeksema & Liang, 2018*), and isolated guitar transcription (*Burlet & Hindle, 2017*). Hence, deep learning is also a fascinating trend in bioinformatics and computational biology research. This study attempts to present a framework to apply deep learning in bioinformatics by using two-dimensional convolutional neural network (2D CNN), which is one popular type of deep neural networks. We anticipate our method will lead to a significant improvement when compared to traditional machine learning techniques in the bioinformatics field.

In earlier years, researchers used shallow neural networks for solving a number of problems in bioinformatics and computational biology. For example, Ou constructed QuickRBF package (*Oyang et al., 2005*) for training radial basis function (RBF) networks and applied them on several bioinformatics problems including classifying electron transport proteins (*Le, Nguyen & Ou, 2017*), transporters (*Le, Sandag & Ou, 2018*), and binding sites (*Le & Ou, 2016a*; *Le & Ou, 2016b*). *Chang & Lin (2011)* introduced LibSVM to help biologists implement bioinformatics models by using support vector machines. Recently, as deep learning has been successfully applied in various fields, researchers started to use it in bioinformatics problems, e.g., prediction of piRNA (*Wang, Hoeksema & Liang, 2018*) and ab initio protein secondary structure (*Spencer, Eickholt & Cheng, 2015*). Although those studies achieved very good performances, we believe that we can obtain superior results by using 2D CNN in some bioinformatics applications. In this study, we applied our architecture in the prediction of SNARE proteins, which is one of the most vital molecules in the life sciences.

SNARE is an evolutionary superfamily of small proteins that have a conservation pattern of 60–70 amino acids (SNAP motifs) in their cytoplasmic domain. SNARE proteins catalyze cell membrane integration in eukaryotes and are essential for a wide range of cellular processes, including cell growth, cytokinesis, and synaptic transmission (*Jahn & Scheller, 2006*; *Wickner & Schekman, 2008*). Most SNAREs contain only one SNARE motif adjacent to a single C-terminal membrane (e.g., synaptobrevin 2 and syntaxin 1). Some SNAREs contain two SNARE motifs that are connected by a long linkage and non-transmembrane sequence (e.g., SNAP-25) but are attached to the membrane through a post-translational modification such as palmitoylation. Various types of SNARE proteins now identified and several studies demonstrated that a functional loss of SNARE proteins has been implicated in numerous diseases (e.g., neurodegenerative (*Hou et al., 2017*), mental illness (*Honer et al., 2002*), cancer (*Meng & Wang, 2015*; *Sun et al., 2016*), and so on). Therefore, SNARE proteins play an important function in the cell and there is a need to develop some bioinformatics techniques to identify them.

Because of the essential role in human diseases, SNARE proteins attracted various researchers who conducted their research on them. For instance, Kloepper team attempted

to build a database to store and classify SNARE proteins (*Kienle, Kloepper & Fasshauer, 2009*; *Kloepper, Kienle & Fasshauer, 2007*; *Kloepper, Kienle & Fasshauer, 2008*). Next, *Van Dijk et al. (2008)* built a framework to predict functions of SNAREs in sub-Golgi localization. Moreover, *Weimbs et al. (1997)* used bioinformatics techniques to analyze conserved domains in SNARE. *Yoshizawa et al. (2006)* extracted sequence motifs and the phylogenetic features of SNARE-dependent membrane trafficking. *Shi et al. (2016)* directed targeting of membrane fusion by SNARE mimicry by convergent evolution of Legionella effectors. *Lu (2015)* analyzed the destructive effect of botulinum neurotoxins on the SNARE protein and proposed that the truncated SNAP-25 mutants will disrupt the assembly of the SNARE core complex, and then inhibit the synaptic membrane fusion accordingly.

Most published works on SNARE proteins achieved high performance, but to our knowledge, no researcher conducted the prediction of SNARE proteins using machine learning techniques. It is challenging and motivates us to create a precise model for this. Besides that, we also applied deep learning in this problem, which is a modern technique for classification and obtain high accuracies in various fields. Based on the advantages of deep learning, this study consequently proposes the use of a 2D convolutional neural network (CNN) constructed from position-specific scoring matrix (PSSM) profiles to identify SNARE proteins. The basic principle has already been successfully applied to identify electron transporting proteins (*Le, Ho & Ou, 2017*) and Rab GTPases (*Le, Ho & Ou, 2018*). Thus, in this paper, we extend this approach to identify the molecular functions of SNARE proteins. The main achievements, including contributions to the field, are presented as follows: (i) development of a deep learning framework to identify SNARE functions from protein sequences, in which our model exhibited a significant improvement beyond traditional machine learning algorithms; (ii) first computational study to identify SNARE proteins and provide useful information to biologists to discover the SNARE molecular functions; (iii) valid benchmark dataset to train and test SNARE proteins with high accuracy, which forms a basis for future research on SNARE proteins.

As shown in a series of recent publications (*Chen et al., 2018*; *Cheng, Xiao & Chou, 2018a*; *Cheng, Xiao & Chou, 2018b*; *Chou, Cheng & Xiao, 2018*; *Feng et al., 2018*; *Jia et al., 2019*; *Khan et al., 2018*; *Xiao et al., 2018b*), to develop a really useful statistical predictor for a biological system, one should observe the guidelines of Chou's 5-step rule (*Chou, 2011*) to make the following five steps very clear: (i) how to construct or select a valid benchmark dataset to train and test the predictor; (ii) how to formulate the statistical samples with an effective mathematical expression that can truly reflect their intrinsic correlation with the target to be predicted; (iii) how to introduce or develop a powerful algorithm (or engine) to operate the prediction; (iv) how to properly perform cross-validation tests to objectively evaluate the anticipated accuracy of the predictor; (v) how to provide source code and dataset that are accessible to the public. Below, we are to describe how to deal with these steps one-by-one.

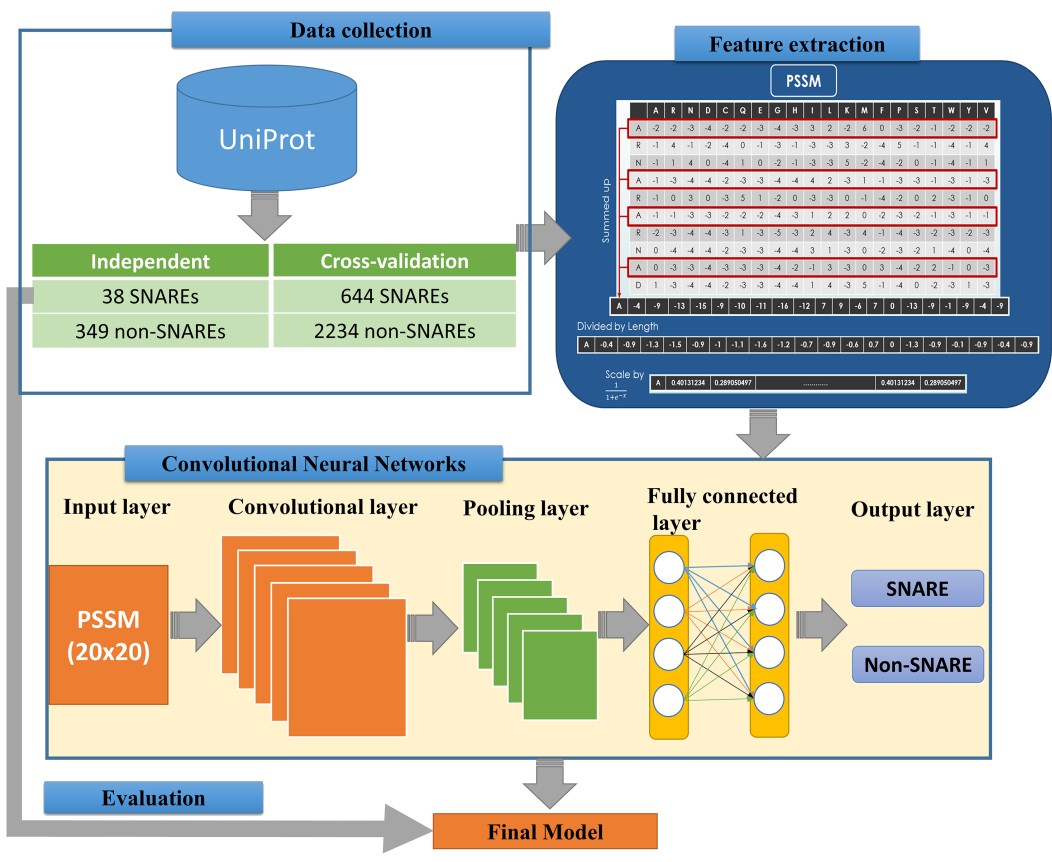

**Figure 1    Flowchart for identifying SNARE proteins using two-dimensional convolutional neural networks.**

# MATERIALS & METHODS

We implemented an efficient framework for identifying SNARE proteins by using a 2D CNN and PSSM profiles. The framework consists of four procedures: data collection, feature extraction, CNN generation, and model evaluation. Figure 1 presents the flowchart of our framework, and its details are described as follows.

## Dataset

The dataset was retrieved from the UniProt database (by 22-10-2018) (*UniProt Consortium, 2014*), which is one of the comprehensive resources for the protein sequence. First of all, we collected all SNAREs proteins from the UniProt annotation (by using keyword "SNARE"). Note that only reviewed proteins (records with information extracted from literature and curator-evaluated computational analysis) were collected. Subsequently, BLAST (*Altschul et al., 1997*) was applied to remove the redundant sequences with similarity more than 30%. However, after this process, the rest of proteins only reached 245 SNAREs, and the number of data points was insufficient for a precise deep learning model. Hence, we used a cut-off level of 100% in the cross-validation dataset for more data to create a significant model.

**Table 1  Statistics of all retrieved SNARE and non-SNARE proteins.**

|  | Cross-validation | Independent |
|---|---|---|
| SNARE | 644 | 38 |
| Non-SNARE | 2,234 | 349 |

We still used similarity of 30% in the independent dataset to evaluate the performance of the model. This step is a very important step to check if the model was overfitting or not.

The proposed problem was the binary classification between SNARE proteins and general proteins, thus we collected a set of general proteins as negative data. In order to create a precise model, there is a need to collect negative dataset which has a similar function and structure with the positive dataset. From that, it is challenging to build a precise model but it increases our contribution to the predictor. It will also help us decrease the number of negative data collected. After considering the structure and function, we chose vesicular transport protein, which is a general protein including SNARE protein. We counted it as negative data to perform the classification problem. We removed the redundant data between two datasets as well as the sequences with similarity more than 30%. Finally, there were 682 SNARE proteins and 2583 non-SNARE proteins used. We then divided data into cross-validation and independent dataset. The detail of the dataset using in this study is listed in Table 1.

## Encoding feature sets from the protein sequence information

In order to convert the protein sequence information into feature sets, we applied the PSSM matrices for FASTA sequences. A PSSM profile is a matrix represented by all motifs in biological sequences in general and in protein sequences in particular. It is created by rendering two sequences having similar structures with different amino acid compositions. Therefore, PSSM profiles have been adopted and used in a number of bioinformatics problems, e.g., prediction of protein secondary structure (*Jones, 1999*), protein disorder (*Shimizu, Hirose & Noguchi, 2007*), and transport protein (*Ou, Chen & Gromiha, 2010*) with significant improvements.

Since the retrieved dataset is in FASTA format, it needs to be converted into PSSM profiles. To perform this task, we used PSI-BLAST (*Altschul et al., 1997*) to search all the sequence alignments of proteins in the non-redundant (NR) database with two iterations. The query to produce the PSSM profile is as follows:

*psiblast.exe -num_iterations 2 -db <nr>-in_msa <fasta_file>-out_ascii_<pssm_file>*

The feature extraction part of Fig. 1 indicates the information of generating the 400 PSSM capabilities from original PSSM profiles. Each amino acid in the sequence is represented by a vector of 20 values (each row). First, we summed up all rows with the same amino acid to transform the original PSSM profiles to PSSM profiles with 400 dimensions. The purpose of this step is to force this data type into something easier for the neural network to deal with. Each element of the 400D input vector was then divided by the sequence length and then be scaled before inserting into neural networks.

## Input layers for 2D convolutional neural networks

The architecture of our CNN is described in the below part of Fig. 1. The CNN contains three layers: an input layer, hidden layers (including convolutional, pooling and fully connected layers), and an output layer. CNN had been applied in numerous applications in various fields and convinced wonderful results (*Amidi et al., 2018*; *Palatnik de Sousa, 2018*). In our study, an input of the CNN is a PSSM corresponding to the protein sequences. We then propose a method to predict SNARE proteins by using their PSSM profiles as the input data. With this type of dataset, we assumed the PSSM profile with $20 \times 20$ matrix as a grayscale image with $20 \times 20$ pixels, we can then train the model with two-dimensional CNN. The input PSSM profile was then connected to our 2D CNN in which we set a variety of parameters to improve the performance of the model. By using a 2D CNN rather than other neural network structures, we aimed to capture as many hidden spatial features as possible in the PSSM matrices. This approach guarantees the correctness of the generated features and prevents the disorder problem inside the amino acid sequences. The more hidden layers generated, the more hidden features generated in CNN to identify SNARE proteins easily. In this work, we used four filter layers (with 32, 64, 128, and 256 filters) and three different kernel sizes in each filter.

## Multiple hidden layers for deep neural networks

Following the input layer, hidden layers aim to generate matrices to learn the features. We established the hidden layers that contained various sub-layers with different parameters and shapes. Those 2D sub-layers are zero padding, convolutional, max pooling and fully-connected layers with different numbers of filters. All of the layers are combined together to become the nodes in the deep neural networks. The quality of our model was determined by the number of layers and parameters. The first layer of our 2D CNN architecture is the zero padding 2D layer, which added zero values at the beginning and the end of $20 \times 20$ matrices. The shape matrix changed to $22 \times 22$ dimensions when we added the zero padding layer into our network. After we applied the filters into the input shape, the output dimension was not different under the effect of the zero padding.

$$zp = \frac{k-1}{2} \tag{1}$$

where $k$ is the filter size. Next, the 2D convolution layer was used with a kernel size of $3 \times 3$, meaning that the features will be learned with the $3 \times 3$ matrices and sliced to the end. After each step, the next layer will take the weights and biases from its previous layer and train again. Normally, a 2D max-pooling layer follows the 2D convolution layer. There are several parameters for a max-pooling layer, i.e., loop size and stride. In our study, we performed max pooling by a stride of 2 through the selection of the maximum value over a window of 22. By using this process, we can reduce the processing time in the next layers. The output size of a convolutional layer is computed as follow.

$$os = \frac{w-k+2p}{s} + 1 \tag{2}$$

where $w$ is the input size, $k$ is the filter size, $p$ is the padding and $s$ is the stride size.

## Output layers

The first layer in the output layer is a flatten layer. A flatten layer is always included before fully connected layers to convert the input matrix into a vector. We applied two fully connected layers in which each node is fully-connected to all the nodes of the previous layer. Fully connected layers are typically used in the last stages of CNNs. All the nodes of the first layer are connected to the flatten layer to allow the model to gain more knowledge and perform better. The second layer connects the first fully-connected layer to the output layer. Moreover, we inserted the next layer, dropout, to enhance the performance results of the model and it also helps our model prevent overfitting (*Srivastava et al., 2014*). In the dropout layer, the model will randomly deactivate a number of neurons with a certain probability p. By tuning the dropout value (from 0 to 1), we will save a lot of computing time for the next layers, and the training time will be faster. Furthermore, an additional non-linear operation called ReLU (Rectified Linear Unit) was performed after each convolution operation. To define the ReLU output, we used this formula:

$$f(x) = \max(0, x) \tag{3}$$

where $x$ is the number of inputs into a neural network. The output of the model was computed through a softmax function by which the probability for each possible output was determined. The softmax function is a logistic function which is defined by the following formula:

$$\sigma(z)_i = \frac{e^{z_i}}{\sum_{k=1}^{K} e^{z_k}} \tag{4}$$

where $z$ is the input vector with K-dimensional vector, K-dimensional vector $\sigma(z)$ is real values in the range (0, 1) and $j^{th}$ class is the predicted probability from sample vector $x$. In summary, we set a total of 233,314 trainable parameters in the model (Table 2).

## Performance evaluation

The most important purpose of the present study was to predict whether or not a sequence is SNARE protein; therefore, we used ''Positive'' to define the SNARE protein, and ''Negative'' to define the non-SNARE protein. For each dataset, we first trained the model by applying 5-fold cross-validation technique on the training dataset. Based on the 5-fold cross-validation results, hyper-parameter optimization process was employed to find the best model for each dataset. Finally, the independent dataset was used to assess the predictive ability of the current model.

Based on the Chou's symbols introduced for studying protein signal peptides (*Chou, 2001*), a set of four intuitive metrics were derived, as given in Eq. 14 of *Chen et al. (2013)* or in Eq. 19 of *Xu et al. (2013)*. For evaluating the performance of the methods, we also adopted Chou's criterion used in many bioinformatics studies (*Chen et al., 2007*; *Feng et al., 2013*; *Taju et al., 2018*). Either the set of traditional metrics copied from math books or the intuitive metrics derived from the Chou's symbols (Eqs. (5)–(8)) are valid only for the single-label systems (where each sample only belongs to one class). For the multi-label systems (where a sample may simultaneously belong to several classes), whose existence

**Table 2 All layers and trainable parameters of the two-dimensional convolutional neural networks in this study.**

| Layer (type) | Output shape | Parameters # |
|---|---|---|
| Zeropadding2d_1 | (None, 3, 22, 20) | 0 |
| Conv2d_1 | (None, 1, 20, 32) | 5,792 |
| Max_pooling2d_1 | (None, 1, 10, 16) | 0 |
| Zeropadding2d_2 | (None, 3, 12, 16) | 0 |
| Conv2d_2 | (None, 1, 10, 64) | 9,280 |
| Max_pooling2d_2 | (None, 1, 5, 32) | 0 |
| Zeropadding2d_3 | (None, 3, 7, 32) | 0 |
| Conv2d_3 | (None, 1, 5, 128) | 36,992 |
| Max_pooling2d_3 | (None, 1, 2, 64) | 0 |
| Zeropadding2d_4 | (None, 3, 4, 64) | 0 |
| Conv2d_4 | (None, 1, 2, 256) | 147,712 |
| Max_pooling2d_4 | (None, 1, 1, 128) | 0 |
| Flatten_1 | (None, 128) | 0 |
| Dense_1 | (None, 256) | 33,024 |
| Dropout_1 | (None, 256) | 0 |
| Dense_2 | (None, 2) | 514 |
| Activation_1 | (None, 2) | 0 |

has become more frequent in system biology (*Cheng, Xiao & Chou, 2017a*; *Cheng, Xiao & Chou, 2017b*; *Cheng et al., 2017a*; *Xiao et al., 2018a*), system medicine (*Cheng et al., 2017b*) and biomedicine (*Qiu et al., 2016*), a completely different set of metrics as defined in (*Chou, 2013*) is absolutely needed. Some standard metrics were used, such as sensitivity, specificity, accuracy, and Matthews correlation coefficient (MCC) using below given formulae (TP, FP, TN, FN are true positive, false positive, true negative, and false negative values, respectively):

$$Sensitivity = 1 - \frac{N_-^+}{N^+}, 0 \leq Sen \leq 1 \tag{5}$$

$$Specificity = 1 - \frac{N_+^-}{N^-}, 0 \leq Spec \leq 1 \tag{6}$$

$$Accuracy = 1 - \frac{N_-^+ + N_+^-}{N^+ + N^-}, 0 \leq Acc \leq 1 \tag{7}$$

$$MCC = \frac{1 - \left(\frac{N_-^+}{N^+} + \frac{N_+^-}{N^-}\right)}{\sqrt{\left(1 + \frac{N_+^- - N_-^+}{N^+}\right)\left(1 + \frac{N_-^+ - N_+^-}{N^-}\right)}}, -1 \leq MCC \leq 1 \tag{8}$$

The relations between these symbols and the symbols in Eqs. (5)–(8) are given by:

$$\begin{cases} N_+^- = FP \\ N_-^+ = FN \\ N^+ = TP + N_-^+ \\ N^- = TN + N_+^- \end{cases} \tag{9}$$

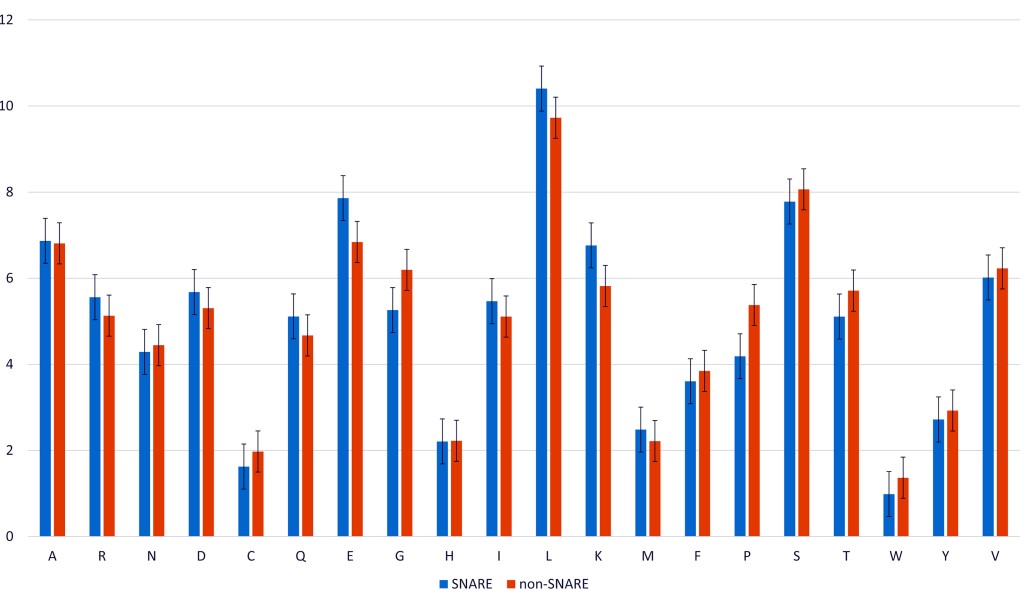

**Figure 2** **Amino acid composition in SNARE and non-SNARE proteins.**

# RESULTS AND DISCUSSIONS

The quality and reliability of the modeling techniques of research is an important factor in the study. Initially, we designed an experiment by analyzing data, perform calculations and take various comparisons in the results and discussions section.

## Composition of amino acid in SNARE and non-SNARE proteins

We analyzed the composition of amino acid and the variance of amino acid composition in SNARE sequences and non-SNARE sequences by computing the frequency between them. Figure 2 illustrates the amino acids which contributed the significantly highest frequency in two different datasets. We realized that the amino acid E, and K, and L occur at the highest frequencies surrounding the SNARE proteins. On the other hand, amino acids G and P occur at the highest frequencies surrounding the non-SNARE proteins. Therefore, these amino acids certainly had an essential role in identifying SNARE proteins. Thus, our model might predict SNARE proteins accurately via the special features from those amino acids contributions.

## Performance for identifying SNARE proteins with 2D CNN

We implemented our 2D CNN architecture by using Keras package with Tensorflow backend. First, we tried to find the optimal setup for the hidden layers by doing experiments using four different filter sizes: 32, 64, 128, and 256. Table 3 demonstrates the performance results from various filter layers in the cross-validation dataset. We easily observe that during the 5-fold cross-validation to identify SNAREs, the model with 256 filters was prominent identifying these sequences with an average 5-fold cross-validation accuracy of 88.2%. The performance results are higher than the performances from the other results

**Table 3  Performance results of identifying SNAREs with different filter layers.**

| Filters | Sens | Spec | Acc | MCC |
|---|---|---|---|---|
| 32 | 68.3 | 91.9 | 86.6 | 0.61 |
| 32–64 | 69.9 | 93.2 | 88 | 0.65 |
| 32–64–128 | 73.3 | 91.5 | 87.5 | 0.64 |
| 32–64–128–256 | 70.5 | 93.3 | 88.2 | 0.65 |

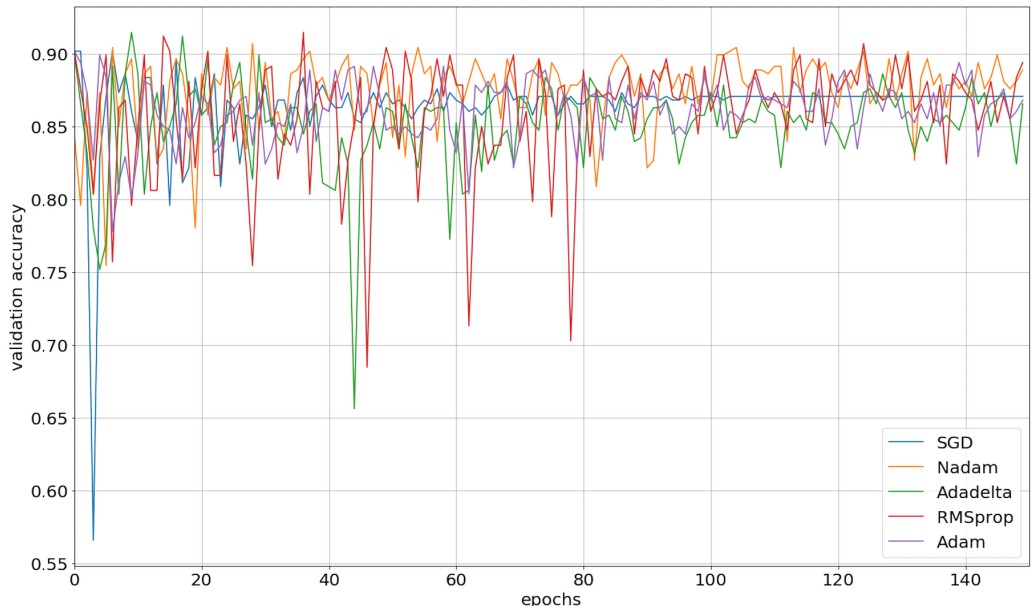

**Figure 3  The validation accuracy on identifying SNARE proteins using different optimizers.**

with other filters. The sensitivity, specificity, and MCC for cross-validation data achieved 70.5%, 93.3%, and 0.65, respectively. Therefore, we used 256 filters for the hidden layer to develop our model. We then optimized the neural networks using a variety of optimizers: rmsprop, adam, nadam, sgd, and adadelta. The model was reinitialized, i.e., a new network is built, after each round of optimization so as to provide a fair comparison between the different optimizers. Overall, the performance results are shown in Fig. 3 and we decided to choose nadam, an optimizer with consistent performance to create our final model.

## Improving the performance results and preventing overfitting problem with dropout

It can be seen that there was a fair difference in performance between using the cross-validation dataset and the independent dataset. It is due to the non-removing similarity in cross-validation, and now we address this issue. To solve this issue, we applied an important technique called dropout (*Srivastava et al., 2014*). Table 4 presents the performances of the model when we varied the dropout value from 0 to 1. It can be seen that the performance from the dropout value of 0.1 was higher than others, with the sensitivity, specificity,

**Table 4 Performance results of identifying SNAREs with different dropout levels.**

| | Cross-validation | | | | Independent | | | |
|---|---|---|---|---|---|---|---|---|
| Dropout | Sens | Spec | Acc | MCC | Sens | Spec | Acc | MCC |
| 0 | 70.5 | 93.3 | 88.2 | 0.65 | 57.9 | 85.7 | 82.9 | 0.33 |
| 0.1 | 72.4 | 94.4 | 89.5 | 0.69 | 44.7 | 95.4 | 90.4 | 0.43 |
| 0.2 | 69.3 | 93.9 | 88.4 | 0.65 | 50 | 87.4 | 83.7 | 0.3 |
| 0.3 | 69.6 | 94.2 | 88.7 | 0.66 | 42.1 | 86 | 81.7 | 0.22 |
| 0.4 | 72 | 92.6 | 88 | 0.65 | 39.5 | 91.4 | 86.3 | 0.29 |
| 0.5 | 69.7 | 94.8 | 89.1 | 0.68 | 36.8 | 92.8 | 87.3 | 0.29 |

accuracy, and MCC of 72.4%, 94.4%, 89.5%, and 0.69, respectively. In the independent dataset, the sensitivity, specificity, accuracy, and MCC were 44.7%, 95.4%, 90.4%, and 0.43, respectively. We can see that the performance of the independent dataset has been already improved and moved closer to that of the cross-validation dataset. Therefore, the overfitting problem was gradually resolved, and we used this dropout value for our final model.

Moreover, the number of epochs used in the experiment extremely affects the performance results. To discover the optimal epoch, we ran our experiments by ranging the epoch value from the first epoch to the 500th epoch. During this process, we saved the checkpoint with the highest performance and used its parameters to create our model. Until this final step, the independent sensitivity, specificity, accuracy, and MCC reached 65.8%, 90.3%, 87.9% and 0.46, respectively. This result is close to that of the cross-validation dataset at the same level of 2D CNN architecture. Finally, our model applied 256 filter layers, nadam optimizer, and dropout value of 0.1 to identify SNARE proteins with the highest performance.

## Comparative performance for identifying SNAREs between 2D CNN and shallow neural networks

We examined the performances of using different machine learning classifiers for identifying SNARE proteins. We used four different classifiers (i.e., nearest neighbor (kNN), Gaussian, Random Forest, and support vector machine (SVM)) to evaluate the model and compared 2D CNN results with their results. For a fair comparison, we definitely used the optimal parameters for all the classifiers in all the experiments. Table 5 shows the performance results between our method and other machine learning algorithms. It can be seen that our 2D CNN exhibited higher performance than those of the other traditional machine learning techniques using the same experimental setup. Especially, our 2D CNN outperformed other algorithms when using the independent dataset.

## Comparative performance for identifying SNAREs between 2D CNN and BLAST search pipeline

To make our prediction have convincing, we aimed to simply BLASTing the SNARE and non-SNARE sequences. The objective of this step is to check whether the first non-identical match was a SNARE/non-SNARE protein. We then compared with our PSSM via PSI-BLAST and the performance results were shown in Table 6. It is easy to say that we are

**Table 5  Comparative performance between 2D CNN and other shallow neural networks.**

| Classifier | Cross-validation | | | | Independent | | | |
|---|---|---|---|---|---|---|---|---|
| | Sens | Spec | Acc | MCC | Sens | Spec | Acc | MCC |
| kNN | 60.1 | 95.4 | 87.5 | 0.62 | 28.9 | 95.1 | 88.6 | 0.28 |
| RandomForest | 59.6 | 98.2 | 89.5 | 0.68 | 15.8 | 98 | 89.9 | 0.23 |
| Gaussian | 93.5 | 30.5 | 44.6 | 0.23 | 81.6 | 23.2 | 28.9 | 0.03 |
| SVM | 35.2 | 98.1 | 84 | 0.48 | 28.9 | 97.1 | 90.4 | 0.34 |
| 2D CNN | 76.6 | 93.5 | 89.7 | 0.7 | 65.8 | 90.3 | 87.9 | 0.46 |

**Table 6  Comparative performance between our classification method and BLAST search pipeline.**

| Method | Cross-validation | | | | Independent | | | |
|---|---|---|---|---|---|---|---|---|
| | Sens | Spec | Acc | MCC | Sens | Spec | Acc | MCC |
| BLAST | 37.0 | 99.3 | 85.3 | 0.53 | 26.3 | 99.4 | 92.2 | 0.44 |
| 2D CNN | 76.6 | 93.5 | 89.7 | 0.7 | 65.8 | 90.3 | 87.9 | 0.46 |

able to reach a better performance when using the PSSM profiles to build a classifier. It also means that BLAST can search a sequence within motifs, but it cannot capture hidden information in sequences. Therefore, it is necessary and useful to create an advanced classifier with stronger features e.g., PSSM profiles in this study.

Furthermore, source codes and publicly accessible web-servers represent the current trend for developing various computational methods (*Chen et al., 2018*; *Cheng, Xiao & Chou, 2018a*; *Cheng, Xiao & Chou, 2018b*; *Chou, Cheng & Xiao, 2018*; *Feng et al., 2018*; *Jia et al., 2019*; *Khan et al., 2018*; *Le, Ho & Ou, 2019*; *Xiao et al., 2018b*). Actually, they have significantly enhanced the impacts of computational biology on medical science (*Chou, 2015*), driving medicinal chemistry into an unprecedented revolution (*Chou, 2017*), here we also publish our source codes and dataset at https://github.com/khanhlee/snare-cnn for presenting the new method reported in this paper.

## CONCLUSIONS

Deep learning, a leading technique in various fields, has been increasingly applied in bioinformatics and computational biology. This study approaches a novel for identifying SNARE proteins by using deep learning. The idea is to transform PSSM profiles into matrices and use them as the input to 2D CNN architectures. We evaluated the performance of our model, which was developed by using a 2D CNN and PSSM profiles, using 5-fold cross-validation and an independent testing dataset. Our method produced superior performance, and compared to other state-of-the-art neural networks, it achieved a significant improvement in all the typical measurement metrics. Using our model, new SNARE proteins can be accurately identified and used for drug development. Moreover, the contribution of this study could help further research to promote the use of 2D CNN in bioinformatics, especially in protein function prediction.

### Funding

The authors received no funding for this work.

### Competing Interests

The authors declare there are no competing interests.

### Author Contributions

- Nguyen Quoc Khanh Le conceived and designed the experiments, performed the experiments, analyzed the data, contributed reagents/materials/analysis tools, prepared figures and/or tables, performed the computation work, authored or reviewed drafts of the paper, approved the final draft.
- Van-Nui Nguyen conceived and designed the experiments, authored or reviewed drafts of the paper, approved the final draft.

### Data Availability

Data is available at GitHub: https://github.com/khanhlee/snare-cnn.

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
