# Peer review of "SNARE-CNN: a 2D convolutional neural network architecture to identify SNARE proteins from high-throughput sequencing data"

_PeerJ Computer Science, doi:10.7717/peerj-cs.177_

## Round 0.1 · original submission · Major Revisions

While the reviewers were thought the manuscript was valuable, a number of concerns were identified. Please try to address the comments provided by both reviewers in your revision.

# Reviewer 1 ·

Basic reporting

See below

Experimental design

See below

Validity of the findings

See below

Additional comments

In this paper, the authors proposed a model to identifying SNARE proteins. SNARE proteins — "SNAP" (Soluble NSF Attachment Protein) REceptor" — are a large protein complex consisting of at least 24 members in yeasts and more than 60 members in mammalian cells. The primary role of SNARE proteins is to mediate vesicle fusion, that is, the fusion of vesicles with their target membrane bound compartments (such as a lysosome). SNAREs are the targets of the bacterial neurotoxins responsible for botulism and tetanus. The authors’ motivation is correct and their efforts should be encouraged. In view of this, the paper holds potential for publication. But to meet the increasingly high quality standard of PeerJ, a careful revision is needed as detailed below.

(1) To make this paper logically more clear, operatively more transparent, and practically more useful, the authors should in the end of the Introduction (or right before the beginning of describing their own method) add a prelude, such as: “As shown in a series of recent publications [1-8], to develop a really useful statistical predictor for a biological system, one should observe the guidelines of Chou’s 5-step rule [9] to make the following five steps very clear: (i) how to construct or select a valid benchmark dataset to train and test the predictor; (ii) how to formulate the statistical samples with an effective mathematical expression that can truly reflect their intrinsic correlation with the target to be predicted; (iii) how to introduce or develop a powerful algorithm (or engine) to operate the prediction; (iv) how to properly perform cross-validation tests to objectively evaluate the anticipated accuracy of the predictor; (v) how to establish a user-friendly web-server for the predictor that is accessible to the public. Below, we are to describe how to deal with these steps one-by-one.” With such a prelude, the outline of this paper and its goal would be crystal clear; its reported results easier for others to repeat. And its attraction to the readership and impact to science would be much higher as well.

(2) It is a big plus to use the intuitive set of metrics (Eqs.5-8) to replace the traditional ones copied from math books as done by many in literature. However, in changing fundamental things, such as metrics or rules, the authors should give more description, such as: “Based on the Chou’s symbols introduced for studying protein signal peptides [10], a set of four intuitive metrics were derived [11, 12], as given in Eq.14 of [11] or in Eq.19 of [12].” Also, the authors should cite as many credible publications [11, 13-23] as possible to justify their decision. Furthermore, it is instructive for the authors to add the following in-depth discussion as saying “Either the set of traditional metrics copied from math books or the intuitive metrics derived from the Chou’s symbols (Eqs.5-8) are valid only for the single-label systems (where each sample only belongs to one class). For the multi-label systems (where a sample may simultaneously belong to several classes), whose existence has become more frequent in system biology [24-27], system medicine [28, 29] and biomedicine [30], a completely different set of metrics as defined in [31] is absolutely needed.”

(3) It is one more big plus that the authors have provided a web-server for their prediction model. To further stress such an advantage, the authors should in the relevant context add a discussion: “User-friendly and publicly accessible web-servers represent the current trend for developing various computational methods [1-8]. Actually they have significantly enhance the impacts of computational biology on medical science [32], driving medicinal chemistry into an unprecedented revolution [33], here we also provide a web-server at https://github.com/khanhlee/snare-cnn for the new method reported in this paper.”

REFERENCES

[1] W. Chen, H. Ding, X. Zhou, H. Lin, iRNA(m6A)-PseDNC: Identifying N6-methyladenosine sites using pseudo dinucleotide composition. Analytical Biochemistry 561-562 (2018) 59-65.
[2] X. Cheng, X. Xiao, pLoc-mEuk: Predict subcellular localization of multi-label eukaryotic proteins by extracting the key GO information into general PseAAC. Genomics 110 (2018) 50-58.
[3] P. Feng, H. Yang, H. Ding, H. Lin, W. Chen, iDNA6mA-PseKNC: Identifying DNA N6-methyladenosine sites by incorporating nucleotide physicochemical properties into PseKNC. Genomics doi:10.1016/j.ygeno.2018.01.005 (2018).
[4] Y.D. Khan, N. Rasool, W. Hussain, S.A. Khan, iPhosT-PseAAC: Identify phosphothreonine sites by incorporating sequence statistical moments into PseAAC. Analytical Biochemistry 550 (2018) 109-116.
[5] X. Xiao⁠, Z.C. Xu⁠, W.R. Qiu⁠, P. Wang, H.T. Ge, iPSW(2L)-PseKNC: A two-layer predictor for identifying promoters and their strength by hybrid features via pseudo K-tuple nucleotide composition. Genomics doi:10.1016/j.ygeno.2018.12.001 (2018).
[6] X. Cheng, X. Xiao, pLoc-mGneg: Predict subcellular localization of Gram-negative bacterial proteins by deep gene ontology learning via general PseAAC. Genomics 110 (2018) 231-239.
[7] K.C. Chou, X. Cheng, X. Xiao, pLoc_bal-mHum: predict subcellular localization of human proteins by PseAAC and quasi-balancing training dataset Genomics doi:10.1016/j.ygeno.2018.08.007 (2018).
[8] J. Jia, X. Li, W. Qiu, X. Xiao, iPPI-PseAAC(CGR): Identify protein-protein interactions by incorporating chaos game representation into PseAAC. Journal of Theoretical Biology 460 (2019) 195-203.
[9] K.C. Chou, Some remarks on protein attribute prediction and pseudo amino acid composition (50th Anniversary Year Review). Journal of Theoretical Biology 273 (2011) 236-247.
[10] K.C. Chou, Prediction of signal peptides using scaled window. Peptides 22 (2001) 1973-1979.
[11] W. Chen, P.M. Feng, H. Lin, iRSpot-PseDNC: identify recombination spots with pseudo dinucleotide composition Nucleic Acids Research 41 (2013) e68.
[12] Y. Xu, X.J. Shao, L.Y. Wu, N.Y. Deng, iSNO-AAPair: incorporating amino acid pairwise coupling into PseAAC for predicting cysteine S-nitrosylation sites in proteins. PeerJ 1 (2013) e171.
[13] H. Lin, E.Z. Deng, H. Ding, W. Chen, iPro54-PseKNC: a sequence-based predictor for identifying sigma-54 promoters in prokaryote with pseudo k-tuple nucleotide composition. Nucleic Acids Research 42 (2014) 12961-12972.
[14] F. Ali, M. Hayat, Classification of membrane protein types using Voting Feature Interval in combination with Chou's Pseudo Amino Acid Composition. J Theor Biol 384 (2015) 78-83.
[15] W. Chen, P. Feng, H. Ding, H. Lin, iRNA-Methyl: Identifying N6-methyladenosine sites using pseudo nucleotide composition. Analytical Biochemistry 490 (2015) 26-33.
[16] J. Jia, Z. Liu, X. Xiao, iPPI-Esml: an ensemble classifier for identifying the interactions of proteins by incorporating their physicochemical properties and wavelet transforms into PseAAC. J Theor Biol 377 (2015) 47-56.
[17] B. Liu, L. Fang, R. Long, X. Lan, iEnhancer-2L: a two-layer predictor for identifying enhancers and their strength by pseudo k-tuple nucleotide composition. Bioinformatics 32 (2016) 362-369.
[18] W. Chen, P. Feng, H. Yang, H. Ding, H. Lin, iRNA-AI: identifying the adenosine to inosine editing sites in RNA sequences. Oncotarget 8 (2017) 4208-4217.
[19] P.K. Meher, T.K. Sahu, V. Saini, A.R. Rao, Predicting antimicrobial peptides with improved accuracy by incorporating the compositional, physico-chemical and structural features into Chou's general PseAAC. Sci Rep 7 (2017) 42362.
[20] P. Feng, H. Ding, H. Yang, W. Chen, H. Lin, iRNA-PseColl: Identifying the occurrence sites of different RNA modifications by incorporating collective effects of nucleotides into PseKNC. Molecular Therapy - Nucleic Acids 7 (2017) 155-163.
[21] W.R. Qiu, B.Q. Sun, X. Xiao, Z.C. Xu, J.H. Jia, iKcr-PseEns: Identify lysine crotonylation sites in histone proteins with pseudo components and ensemble classifier. Genomics 110 (2018) 239-246.
[22] M. Arif, M. Hayat, Z. Jan, iMem-2LSAAC: A two-level model for discrimination of membrane proteins and their types by extending the notion of SAAC into Chou's pseudo amino acid composition. J Theor Biol 442 (2018) 11-21.
[23] M.S. Krishnan, Using Chou's general PseAAC to analyze the evolutionary relationship of receptor associated proteins (RAP) with various folding patterns of protein domains. J Theor Biol 445 (2018) 62-74.
[24] X. Cheng, X. Xiao, pLoc-mPlant: predict subcellular localization of multi-location plant proteins via incorporating the optimal GO information into general PseAAC. Molecular BioSystems 13 (2017) 1722-1727.
[25] X. Cheng, X. Xiao, pLoc-mVirus: predict subcellular localization of multi-location virus proteins via incorporating the optimal GO information into general PseAAC. Gene (Erratum: ibid., 2018, Vol.644, 156-156) 628 (2017) 315-321.
[26] X. Xiao, X. Cheng, G. Chen, Q. Mao, pLoc_bal-mGpos: predict subcellular localization of Gram-positive bacterial proteins by quasi-balancing training dataset and PseAAC. Genomics doi:10.1016/j.ygeno.2018.05.017 (2018).
[27] X. Cheng, S.G. Zhao, W.Z. Lin, X. Xiao, pLoc-mAnimal: predict subcellular localization of animal proteins with both single and multiple sites. Bioinformatics 33 (2017) 3524-3531.
[28] X. Cheng, S.G. Zhao, X. Xiao, iATC-mISF: a multi-label classifier for predicting the classes of anatomical therapeutic chemicals. Bioinformatics (Corrigendum, ibid., 2017, Vol.33, 2610) 33 (2017) 341-346.
[29] X. Cheng, S.G. Zhao, X. Xiao, iATC-mHyb: a hybrid multi-label classifier for predicting the classification of anatomical therapeutic chemicals. Oncotarget 8 (2017) 58494-58503.
[30] W.R. Qiu, B.Q. Sun, X. Xiao, Z.C. Xu, iPTM-mLys: identifying multiple lysine PTM sites and their different types. Bioinformatics 32 (2016) 3116-3123.
[31] K.C. Chou, Some remarks on predicting multi-label attributes in molecular biosystems. Molecular Biosystems 9 (2013) 1092-1100.
[32] K.C. Chou, Impacts of bioinformatics to medicinal chemistry. Medicinal Chemistry 11 (2015) 218-234.
[33] K.C. Chou, An unprecedented revolution in medicinal chemistry driven by the progress of biological science. Current Topics in Medicinal Chemistry 17 (2017) 2337-2358.

Reviewer 2 ·

Basic reporting

The paper has some typos and difficult to follow sections. I’ve tried to highlight some of the more confusing sections in the following comments.

Minor Points:

Typo in abstract (76.6%%, should be 76.6%)

Experimental design

It isn’t clear how cross-validation and independent data sets were produced. As far as I can tell, Uniprot was searched for all SNARE proteins and then proteins that shared over 30% similarity were removed, leaving 245 SNARE proteins. Figure 1, seems to indicate that there are 682 SNAREs used to build and test the model. Where were the extra SNARE proteins found? Is the 682 number found without any similarity filtering?


Assuming I’ve read the dataset section correctly, all of the non-SNARE proteins tested are classified as vesicular transport proteins. I’m concerned that the model is thus trained to only differentiate between SNARE proteins and VT proteins. If you put a random collection of protein sequences through the network, does the model always predict non-SNARE?


I’m also confused as to how PSI-BLAST was used to produce the PSSM profiles. It appears that a PSSM is built for each protein used to train and test the model. If you used the command line version of PSI-BLAST, can you provide the script or command used to produce the profiles? Also, I don’t understand the part of Figure 1, where all of the A rows are summed up. What is represented in the rows in that image?

Validity of the findings

No comments, beyond the previous sections.

Additional comments

I think the most interesting part of this paper is adapting the convolutional NN type for PSSM profiles. Treating the PSSMs as essentially grayscale images is an interesting if unexpected attempt to force this data type into something easier for the neural network to deal with.

Given that one of the goals of the study is to build a model to hopefully find new SNARE proteins (presumably in newly sequenced organisms), I’m a bit surprised that you didn’t make the comparison to simply BLASTing the SNARE and non-SNARE sequences. The output could then be assessing as to whether the first non-identical match was a SNARE/non-SNARE protein. I understand if the true goal was to only assess modern classification methods, but given that PSI-BLAST is already in the pipeline for building the PSSMs, this would seem to be a relevant comparison.

---

## Round 0.2 · Minor Revisions

Assuming that the web-server is not actually available, please implement the change suggested by the reviewer.

Reviewer 2 ·

Basic reporting

No comments

Experimental design

No comments

Validity of the findings

No comments

Additional comments

I'm satisfied with the changes made by the authors and they address all my major points.

I would revise the end of the introduction to either exclude or modify the user-friendly web-server suggestion. As far as I can tell, there is no such user-friendly webserver where a potential user can submit a protein sequence for SNARE vs non-SNARE prediction. I don't think this is a particular problem though as I don't think all methods need to have an associated user-friendly server and that the code shared by the authors is a decent start. I know this is in direct conflict with the other reviewer's suggestion, but including this section in the introduction makes it appear that such a server has been provided.

---

## Round 0.3 · accepted · Accept

Thank you for addressing the reviewer comments and congratulations again!